# Balancing Fidelity and Diversity in Diffusion Models via Symmetric Attention Decomposition: Hopfield Perspective

Hyunmin Cho [1] Woo Kyoung Han [1] Kyong Hwan Jin [1 ‡]

## Abstract

We characterize the pre-softmax attention matrix $\mathbf{Q}\mathbf{K}^\top$ in transformers as an associative memory matrix encoding pairwise associations between input features. By decomposing this matrix into its symmetric and skew-symmetric parts, we interpret the symmetric component as governing the structure of the *energy landscape*, and the skew-symmetric component as driving *circulation* on that landscape. Leveraging the energy formulation induced by the symmetric component, we derive Hopfield-style stability measures that quantify the stability of retrieved features. We observe meaningful correlations between Hopfield-style stability measures and the fidelity–diversity trade-offs in generation. Finally, we propose a controllable knob to modulate this trade-off by modifying the circulation of the underlying dynamics. Code is available at our Project Page ⬡.

## 1. Introduction

Diffusion models (Ho et al., 2020; Rombach et al., 2022; Podell et al., 2024; Esser et al., 2024; Labs et al., 2025) have become a leading paradigm for image generation. Their success is largely driven by the attention mechanism (Vaswani et al., 2017), which enables the integration of global context and long-range dependency modeling throughout the denoising process (Nichol & Dhariwal, 2021). While this global connectivity facilitates richer compositional associations that enhance novelty and variety (Zhang et al., 2019), it is simultaneously prone to causing spurious mixing of incompatible features, such as the blending of materials between two distinct objects (Oriyad et al., 2025). Crucially, distinguishing between such beneficial context integration and harmful semantic leakage remains non-trivial, as they

---

[1]Department of Electrical Engineering, Korea University, Seoul, South Korea. Correspondence to: Kyong Hwan Jin <kyong_jin@korea.ac.kr>.

*Proceedings of the 43rd International Conference on Machine Learning*, Seoul, South Korea. PMLR 306, 2026. Copyright 2026 by the author(s).

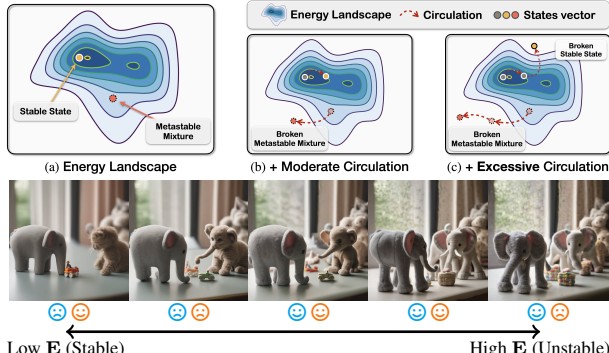

*Figure 1.* **Skew perturbation and the fidelity–diversity trade-off. Top**: We decompose $\mathbf{Q}\mathbf{K}^\top$ into symmetric (energy) and skew (circulation) parts. (a) The symmetric part gives stable but low-diversity retrieval. (b) Moderate skew perturbation breaks metastable mixtures while preserving stable states. (c) Excessive perturbation destabilizes even well-formed retrievals, producing artifacts. **Bottom**: Moderate skew perturbation improves diversity, but excessive perturbation causes hallucinations. ☺/☹ denote positive/negative *diversity*; ☺/☹ denote positive/negative *fidelity*.

share the same underlying mechanism. To address this ambiguity, our goal is to (i) *identify* when attention settles into spurious mixtures, and (ii) *control* this behavior to navigate the trade-off between coherent structure and diversity.

The perspective of associative memory provides a principled lens on these challenges (Amari, 1972; Nakano, 1972; Little, 1974; Hopfield, 1982). Recent dense associative memory work further suggests that the choice of energy function can substantially reshape the landscape of local minima, even giving rise to additional emergent memories beyond stored patterns (Hoover et al., 2026). Building on the insight that transformer self-attention approximates the update rule of a modern Hopfield network (Ramsauer et al., 2021), we re-frame spurious mixing as entrapment in metastable states (local energy minima where the model settles on an incoherent combination of distinct patterns). However, standard analyses typically operate at a *token-wise* level, treating attention merely as a retrieval mechanism. This token-centric view restricts the capture of the rich interaction dynamics encoded in the attention matrix itself.

Furthermore, these interpretations often overlook the dynamical consequences of asymmetric association matrices. In recurrent associative memories, such asymmetry is known

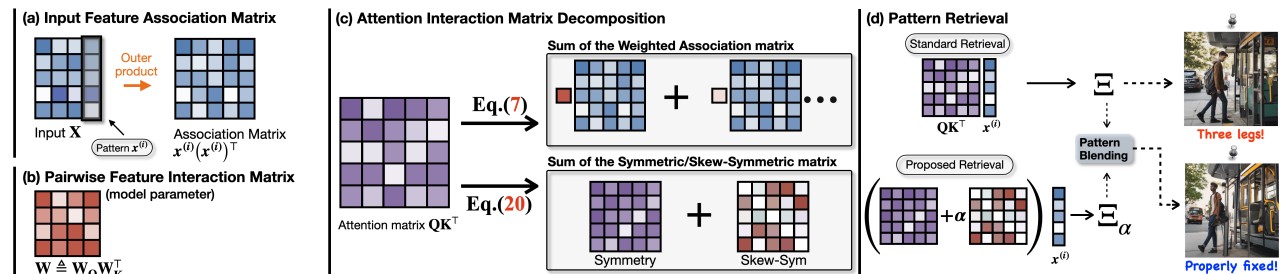

*Figure 2.* **Associative memory framework encoding pairwise feature interactions and its decomposition. (a)–(b)** We characterize the attention mechanism as an associative memory encoding *pairwise feature interactions*. (a) Viewing input features $\mathbf{X} \in \mathbb{R}^{L \times d_{\text{in}}}$ as a set of features $\boldsymbol{x}^{(i)} \in \mathbb{R}^L$, (b) the learned interaction matrix $\mathbf{W}$ encodes the association strength between these feature pairs. **(c)** The resulting attention matrix $\mathbf{Q}\mathbf{K}^\top$ can be decomposed into a *symmetric component* and a *skew-symmetric component*. The symmetric term defines a static *energy landscape* governing the stability of retrieved features, while the skew-symmetric term drives *circulation*, acting as a directional force. **(d)** Control via Circulation. Standard retrieval often settles into *metastable mixtures* (e.g., the incoherent 'Three legs' structure). We propose using the skew-symmetric component as a controllable knob. Amplifying this component injects circulation-driven drift to *perturb* the metastable state, *restoring structural coherence*.

to reshape the attractor structure, allowing non-fixed-point attractors such as limit cycles (Hwang et al., 2019). This structural property is crucial, as it induces circulation that helps perturb and *destabilize* metastable mixtures (Singh et al., 1995; Chengxiang et al., 2000).

In this work, we characterize the attention matrix $\mathbf{Q}\mathbf{K}^\top$ as a dynamic associative memory that encodes pairwise feature associations (Figure 2). Unlike prior token-level analyses, our view exposes the association structure that governs the mixing dynamics. Concretely, we decompose $\mathbf{Q}\mathbf{K}^\top$ into a symmetric and a skew component: the symmetric component defines a Hopfield-style *energy landscape*. In contrast, the skew-symmetric component drives *circulation*, acting as a directional force to perturb metastable states (Figure 1). This decomposition reveals that generation quality hinges on the balance between energy-based stability and circulation-driven dynamics. Leveraging this insight, we derive Hopfield-style stability measures, enabling us to *identify* metastable mixtures (Goal (i)). Finally, we exploit the skew-symmetric circulation as a tunable knob to *control* the retrieval process, facilitating the perturbation of metastable mixtures (Goal (ii)). To summarize our contributions:

- We establish an associative memory framework that encodes pairwise feature associations for the attention matrix and introduce a symmetric/skew-symmetric decomposition that disentangles energy-based stability from circulation-driven drift.

- Leveraging the symmetric component, we derive Hopfield-style stability measures that quantify the stability of retrieved features, demonstrating their correlation with the fidelity–diversity trade-off (Table 1).

- We propose the skew-symmetric component as a controllable 'circulation knob' for test-time intervention, which injects directional drift to perturb metastable mixtures and restore structural coherence (Table 3).

## 2. Related Work

**Denoising diffusion models** generate samples by learning to invert a progressive noising process, initially introduced in Sohl-Dickstein et al. (2015) and popularized as DDPMs in Ho et al. (2020). Subsequent formulations unify diffusion with score matching and continuous-time SDE/ODE views (Song & Ermon, 2019; Song et al., 2021), and related continuous-time objectives such as flow matching regress vector fields that transport noise to data (Lipman et al., 2023; Liu et al., 2023). Complementing these algorithmic formulations, recent theoretical works have reinterpreted these generative dynamics through the lens of associative memory, analyzing how diffusion trajectories disperse information and balance memorization with generalization (Ambrogioni, 2023; Hoover et al., 2023; Pham et al., 2025).

**Associative memory networks** are grounded in the classical Hopfield network, which defines an energy landscape over binary states. In these models, the system evolves to minimize energy based on the local field inputs (Amari, 1972; Nakano, 1972; Little, 1974; Hopfield, 1982). To overcome the storage limitations inherent to these classical pairwise-interaction models, Krotov & Hopfield (2016) introduced Dense Associative Memories (DAMs), which generalize the energy function by replacing the quadratic interaction term with a rapidly growing nonlinear function (e.g., polynomial or exponential) defined over the stored patterns. The gradient of this energy governs the update dynamics, resulting in sharper basins of attraction and a significantly higher storage capacity.

**Asymmetric associative memories** extend classical associative memory models beyond symmetric couplings by allowing directed interactions between stored states. Whereas symmetric Hopfield-type memories admit an energy-based interpretation with detailed balance, asymmetric interactions break this reversibility and can substantially alter retrieval dynamics (Peretto, 1984; Derrida et al., 1987; Chengxiang

et al., 2000). In the Hopfield model with random asymmetric interactions, the synaptic matrix $\mathbf{J}$ is decomposed into symmetric and asymmetric components:

$$\mathbf{J}_{ij} = \mathbf{J}_{ij}^{\mathrm{s}} + k\,\mathbf{J}_{ij}^{\mathrm{as}} \quad (i \neq j), \qquad \mathbf{J}_{ij}^{\mathrm{as}} = -\mathbf{J}_{ji}^{\mathrm{as}}, \quad (1)$$

where the symmetric part $\mathbf{J}_{ij}^{\mathrm{s}}$ is Hebbian and the skew-symmetric part $\mathbf{J}_{ij}^{\mathrm{as}}$ introduces asymmetry. Singh et al. (1995) analytically counted attractors in this setting and reported that adding an asymmetric component causes an exponential decrease in the total number of attractors, suggesting a mechanism for suppressing metastable states while preserving retrieval when the asymmetry is modest.

**Attention mechanisms** model interactions among a sequence of feature representations and have become a central building block of modern neural architectures (Vaswani et al., 2017). In language models, sequence positions typically correspond to text tokens (Brown et al., 2020; Touvron et al., 2023), whereas in vision-generative backbones they often correspond to image patches, or flattened latent positions. Attention explicitly parameterizes interactions among these positions, making it a natural target for controlling generation behavior through architectural design or inference-time modulation (Chen et al., 2024; Hong, 2024; Kim & Sim, 2025). These studies suggest that attention can serve as a handle for modulating generation dynamics.

**Viewing attention as associative retrieval** bridges memory-based dynamics and transformer attention (Vaswani et al., 2017). Ramsauer et al. (2021) formalize self-attention as a retrieval step in a continuous-state modern Hopfield network, where softmax implements an exponential Gibbs weighting over stored patterns. From a dynamical perspective, D'Amico & Negri (2024) reinterpret self-attention through an energy-based lens, emphasizing attractor-like behavior induced by attention updates. Complementing these activation-centric views, Bietti et al. (2023) offers a parameter-centric perspective, interpreting transformer *weight matrices* as associative memories that store embedding pairs as weighted outer products. However, these connections are typically framed either as token-level retrieval dynamics (Ramsauer et al., 2021; D'Amico & Negri, 2024) or as static memories residing in the parameters (Bietti et al., 2023). Consequently, the role of the underlying *feature interactions* instantiated in the $\mathbf{QK}^\top$ remains underexplored.

## 3. Hopfield Interpretation of Attention Matrix

To analyze the internal structure of attention (Vaswani et al., 2017), we view the input feature map $\mathbf{X} \in \mathbb{R}^{L \times d_{\mathrm{in}}}$ as a collection of $d_{\mathrm{in}}$ *real-valued* features, denoted by

$$\boldsymbol{x}^{(i)} \triangleq [\mathbf{X}]_{:,i} \in \mathbb{R}^L, \qquad i = 1, \ldots, d_{\mathrm{in}}. \quad (2)$$

Let the query and key projections be

$$\mathbf{Q} \triangleq \mathbf{X}\mathbf{W}_Q, \qquad \mathbf{K} \triangleq \mathbf{X}\mathbf{W}_K, \quad (3)$$

where $\mathbf{W}_Q, \mathbf{W}_K \in \mathbb{R}^{d_{\mathrm{in}} \times d_k}$. The pre-softmax attention matrix $\mathbf{QK}^\top$ is then

$$\mathbf{QK}^\top = \mathbf{X}\mathbf{W}_Q\mathbf{W}_K^\top\mathbf{X}^\top. \quad (4)$$

For notational convenience, define the interaction weight matrix

$$\mathbf{W} \triangleq \mathbf{W}_Q\mathbf{W}_K^\top \in \mathbb{R}^{d_{\mathrm{in}} \times d_{\mathrm{in}}}, \quad (5)$$

so that the attention matrix admits the compact factorization

$$\mathbf{QK}^\top = \mathbf{X}\mathbf{W}\mathbf{X}^\top. \quad (6)$$

This expansion shows that $\mathbf{QK}^\top$ is a weighted superposition of rank-one outer products, analogous in form to classical Hopfield-style constructions (Personnaz et al., 1986):

$$\mathbf{QK}^\top = \sum_i^{d_{\mathrm{in}}} \underbrace{W_{ii}\,\boldsymbol{x}^{(i)}\big(\boldsymbol{x}^{(i)}\big)^\top}_{\text{self association}} + \sum_{i \neq j}^{d_{\mathrm{in}}} \underbrace{W_{ij}\,\boldsymbol{x}^{(i)}\big(\boldsymbol{x}^{(j)}\big)^\top}_{\text{hetero association}}. \quad (7)$$

This formulation establishes $\mathbf{QK}^\top$ as an associative memory encoding pairwise feature interactions, dynamically constructed from $\mathbf{X}$ as a weighted superposition of *self-association* and *hetero-association* terms (Figure 2c), with interaction strengths governed by the coefficient $W_{ij}$.

**Hopfield retrieval dynamics.** Given the attention matrix defined in Equation (8) as

$$\mathbf{M}(\mathbf{X}) \triangleq \mathbf{QK}^\top = \mathbf{X}\mathbf{W}\mathbf{X}^\top \in \mathbb{R}^{L \times L}, \quad (8)$$

and for each index $a \in \{1, \ldots, L\}$, the *local field* corresponds to the $a$-th row slice of $\mathbf{M}(\mathbf{X})$, viewed as a column vector:

$$\boldsymbol{m}_a(\mathbf{X}) \triangleq [\mathbf{M}(\mathbf{X})]_{a,:}^\top \in \mathbb{R}^L. \quad (9)$$

Since the local field vectors $\boldsymbol{m}_a(\mathbf{X})$ are real-valued and generally unbounded, we apply a normalization that (i) produces nonnegative, unit-sum mixing weights for *retrieval* and (ii) preserves the ranking induced by the local field. Accordingly, we map each local field vector $\boldsymbol{m}_a(\mathbf{X})$ to simplex-valued coefficients via $\phi : \mathbb{R}^L \to \boldsymbol{\Delta}^{L-1}$, where $\mathbf{1} \in \mathbb{R}^L$ denotes the all-ones vector and

$$\boldsymbol{\Delta}^{L-1} \triangleq \left\{ \boldsymbol{\kappa} \in \mathbb{R}^L : \boldsymbol{\kappa} \geq 0, \ \mathbf{1}^\top\boldsymbol{\kappa} = 1 \right\}, \quad (10)$$

yielding a normalized weighting over the $L$ spatial positions. In the spirit of classical Hopfield retrieval (Hopfield, 1982), we further require $\phi$ to be monotone with respect to the local field: for any $\boldsymbol{m} \in \mathbb{R}^L$ and any $j, k$,

$$[\boldsymbol{m}]_j \geq [\boldsymbol{m}]_k \implies \big[\phi(\boldsymbol{m})\big]_j \geq \big[\phi(\boldsymbol{m})\big]_k. \quad (11)$$

which ensures that such normalization does not alter the preference ordering established by the energy landscape.

We extend $\phi$ row-wise to the matrix operator $\Phi : \mathbb{R}^{L \times L} \to \mathbb{R}^{L \times L}$ for any reference matrix $\mathbf{A} \in \mathbb{R}^{L \times L}$ via

$$\big[\Phi(\mathbf{A})\big]_{a,:} \triangleq \phi\big(\mathbf{A}_{a,:}^\top\big)^\top, \quad \text{for all } a \in \{1, \ldots, L\}, \quad (12)$$

and define the *Hopfield retrieval operator* (Ramsauer et al., 2021)

$$\mathbf{H_X} \triangleq \Phi\big(\mathbf{M(X)}\big) = \Phi\big(\mathbf{XWX^\top}\big). \qquad (13)$$

The retrieved features are then obtained by mixing input features according to $\mathbf{H_X}$:

$$\Xi \triangleq \mathbf{H_X}\,\mathbf{X} \in \mathbb{R}^{L \times d_{\text{in}}}, \quad \xi^{(i)} \triangleq [\Xi]_{:,i}. \qquad (14)$$

**Interpreting self-attention as Hopfield retrieval.** A particular choice of $\Phi$ recovers the standard self-attention retrieval. In particular, with row-wise softmax,

$$\mathbf{H_X} \triangleq \text{softmax}\big(\mathbf{M(X)}\big), \qquad (15)$$

the retrieved features $\Xi$ become

$$\Xi \triangleq \mathbf{H_X X} = \text{softmax}\big(\mathbf{XWX^\top}\big)\mathbf{X}. \qquad (16)$$

Applying a value projection $\mathbf{W}_V \in \mathbb{R}^{d_{\text{in}} \times d_k}$ to the retrieved feature $\Xi$ transforms the mixture into the output representation, yielding the standard update:

$$\text{Attn}(\mathbf{X}) = \Xi\,\mathbf{W}_V. \qquad (17)$$

## 4. Energy-based Stability Measures

Under a Hopfield-style lens, the attention mechanism can exhibit *metastable states* that are not captured by analyses that treat the attention matrix as symmetric, since $\mathbf{QK}^\top$ is generally asymmetric. To disentangle these effects, we decompose $\mathbf{QK}^\top$ into symmetric and skew components.

**Decomposition of attention matrix.** We begin by decomposing the attention matrix into symmetric and skew-symmetric components:

$$\mathbf{QK}^\top = \mathbf{M}_{\text{sym}}(\mathbf{X}) + \mathbf{M}_{\text{skew}}(\mathbf{X}), \text{ where}$$

$$\mathbf{M}_{\text{sym}}(\mathbf{X}) \triangleq \frac{\mathbf{QK}^\top + (\mathbf{QK}^\top)^\top}{2}, \quad \mathbf{M}_{\text{skew}}(\mathbf{X}) \triangleq \frac{\mathbf{QK}^\top - (\mathbf{QK}^\top)^\top}{2}. \qquad (18)$$

Equivalently, it suffices to decompose the learned interaction weight matrix $\mathbf{W}$ as:

$$\mathbf{S} \triangleq \frac{\mathbf{W} + \mathbf{W}^\top}{2}, \quad \mathbf{N} \triangleq \frac{\mathbf{W} - \mathbf{W}^\top}{2}. \qquad (19)$$

Substituting Equation (19) into Equation (6) yields the induced decomposition of the associative memory structure:

$$\mathbf{QK}^\top = \mathbf{XWX}^\top = \overbrace{\mathbf{XSX}^\top}^{\triangleq\,\mathbf{M}_{\text{sym}}(\mathbf{X})} + \overbrace{\mathbf{XNX}^\top}^{\triangleq\,\mathbf{M}_{\text{skew}}(\mathbf{X})}. \qquad (20)$$

This decomposition allows us to separately analyze how the symmetric and skew components of the attention matrix contribute to the denoising process. Figure 3 qualitatively illustrates this separation: the symmetric component preserves global object-level structure, while the skew component captures fine-grained irregular details.

**Energy of attention matrix.** Since $\mathbf{M}_{\text{sym}}(\mathbf{X})$ is symmetric, it defines a valid Hopfield-style energy of features. For a *real-valued* feature $\xi \in \mathbb{R}^L$, we define the quadratic energy (Hopfield, 1982; Amit et al., 1985) induced by the

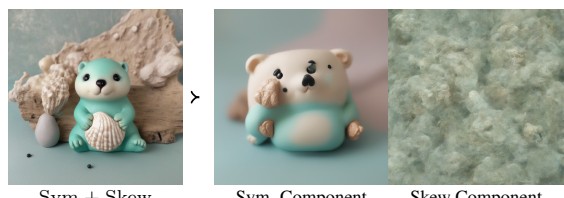

| Sym + Skew | Sym. Component | Skew Component |

*Figure 3.* **Visualization of samples generated via decomposed components.** Samples generated through the sym. component encapsulate the underlying global structure, whereas those generated via the Skew component manifest fine-grained, irregular details.

symmetric component as:

$$E_\mathbf{X}(\xi) \triangleq -\frac{1}{2}\,\xi^\top \mathbf{M}_{\text{sym}}(\mathbf{X})\,\xi. \qquad (21)$$

Lower energy (i.e., more negative $E_\mathbf{X}$) corresponds to a feature $\xi$ that is more *strongly supported* by the associative memory constructed from $\mathbf{X}$ and the learned symmetric interaction rule $\mathbf{S}$.[1]

In contrast, $\mathbf{M}_{\text{skew}}(\mathbf{X})$ is skew-symmetric and therefore contributes *no quadratic energy* for real-valued states:

$$\xi^\top \mathbf{M}_{\text{skew}}(\mathbf{X})\,\xi = (\mathbf{X}^\top\xi)^\top \mathbf{N}(\mathbf{X}^\top\xi) = 0 \qquad (22)$$

since $\mathbf{N} = -\mathbf{N}^\top$ implies

$$\mathbf{u}^\top \mathbf{N}\mathbf{u} = 0, \quad \forall \mathbf{u} \in \mathbb{R}^{d_{\text{in}}}. \qquad (23)$$

Hence, the skew-symmetric component serves to drive the circulation dynamics.

### 4.1. From Global Energy to Local Stability

Equation (21) provides a global measure quantifying how strongly a state $\xi$ is supported by the symmetric interaction component $\mathbf{M}_{\text{sym}}(\mathbf{X})$. However, identifying metastable mixtures requires pinpointing *where* structural incoherence manifests across the $L$ spatial positions; a single scalar energy is insufficient for this purpose.

We therefore complement the global energy with *local stability measures*. These metrics analyze the alignment between the state $\xi$ and its driving local field, thereby exposing the localized conflicts that underlie metastability.

**Local field and local stability.** Importantly, the symmetric component $\mathbf{M}_{\text{sym}}(\mathbf{X})$ is itself a *weighted superposition* of rank-one feature associations,

$$\mathbf{M}_{\text{sym}}(\mathbf{X}) = \sum_{i=1}^{d_{\text{in}}}\sum_{j=1}^{d_{\text{in}}} S_{ij}\,\boldsymbol{x}^{(i)}\big(\boldsymbol{x}^{(j)}\big)^\top, \qquad (24)$$

so its effect on a state $\xi$ is mediated by the induced symmetric local field

$$\boldsymbol{h}_\mathbf{X}(\xi) \triangleq \mathbf{M}_{\text{sym}}(\mathbf{X})\,\xi \in \mathbb{R}^L,$$
$$= \sum_{i=1}^{d_{\text{in}}}\sum_{j=1}^{d_{\text{in}}} S_{ij}\,\boldsymbol{x}^{(i)}\big\langle\boldsymbol{x}^{(j)}, \xi\big\rangle. \qquad (25)$$

---

[1]For notational clarity, we omit the $\sqrt{d_k}$ scaling in $\mathbf{QK}^\top$, which can be absorbed into $\mathbf{W}$ as an overall multiplicative factor.

*Table 1.* **Correlation between evaluation metrics and stability measures.** We report Spearman Rank correlation $\rho$ between sample evaluation metrics (set: **A**) and three Hopfield-style stability measures computed from attention retrieval at each stage (set: **B**). Specifically, for each generated sample, we correlate the final external metric score against the internal stability values averaged over the retrieved features $\boldsymbol{\xi}$ within the specified block range. �usuindicates that higher metric values co-occur with higher stability, while ▯ indicates an association with increased conflict or misalignment. SDXL UNet$_{[s-e]}$ denotes the layer range (s: start, e: end).

| Correlation $\rho$ btw A and B | **B:** | **Down (SDXL UNet$_{[0-47]}$)** | | | **Mid (SDXL UNet$_{[48-67]}$)** | | | **Up (SDXL UNet$_{[68-139]}$)** | | | **All** | | |
|---|---|---|---|---|---|---|---|---|---|---|---|---|---|
| | | $-E_{\mathbf{X}}$ | $r_{\mathbf{X}}$ | $\mathbf{Align_X}$ | $-E_{\mathbf{X}}$ | $r_{\mathbf{X}}$ | $\mathbf{Align_X}$ | $-E_{\mathbf{X}}$ | $r_{\mathbf{X}}$ | $\mathbf{Align_X}$ | $-E_{\mathbf{X}}$ | $r_{\mathbf{X}}$ | $\mathbf{Align_X}$ |
| **A:** Aesthetic Score | | +0.181 | -0.162 | +0.151 | +0.207 | -0.229 | +0.204 | +0.255 | -0.255 | +0.280 | +0.265 | -0.273 | +0.296 |
| LPIPS Diversity | | -0.074 | +0.192 | -0.194 | -0.336 | +0.283 | -0.250 | -0.270 | +0.237 | -0.238 | -0.279 | +0.283 | -0.297 |
| CLIPScore | | +0.040 | +0.155 | -0.202 | -0.158 | +0.088 | -0.042 | -0.006 | -0.073 | +0.142 | -0.010 | +0.030 | -0.014 |
| ImageReward | | +0.129 | -0.161 | +0.146 | -0.168 | +0.102 | -0.090 | -0.122 | +0.114 | -0.192 | -0.074 | +0.046 | -0.074 |

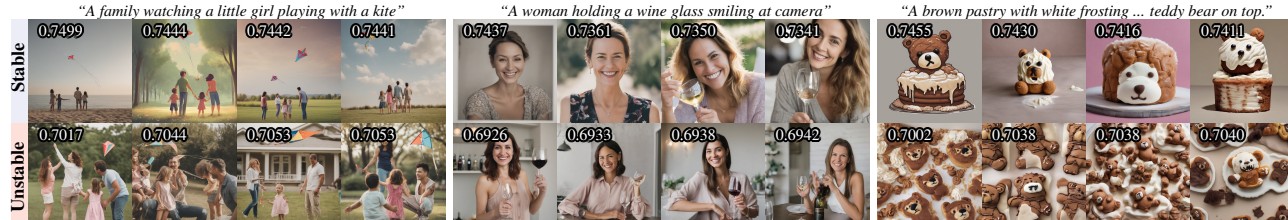

*Figure 4.* **Qualitative comparison from samples sorted by Alignment Score.** For three prompts, we group baseline generations into *Stable* (top row) and *Unstable* (bottom row) subsets according to $\mathbf{Align_X}$. Stable samples show coherent, object-centric structures, whereas unstable samples exhibit diverse but less coherent mixtures. White labels indicate the corresponding $\mathbf{Align_X}$ values.

This expansion explicitly characterizes the mixing mechanism: the field $\boldsymbol{h}_{\mathbf{X}}(\xi)$ is generally a mixture of input feature $\{\boldsymbol{x}^{(i)}\}_{i=1}^{d_{\text{in}}}$, with weights determined by both the symmetric interaction coefficients $S_{ij}$ and the alignment $\langle \boldsymbol{x}^{(j)}, \xi \rangle$. In other words, the response of a retrieved feature is determined by a *superposition of feature* associations supported by the symmetric component. A consistent superposition reinforces the current state across spatial locations, whereas incompatible associations produce coordinate-wise conflicts.

To pinpoint *where* this mixing manifests across the $L$ spatial positions, we measure the *coordinate-wise agreement* between the current state $\xi$ and its driving field $\boldsymbol{h}_{\mathbf{X}}(\xi)$:

$$\boldsymbol{\lambda}_{\mathbf{X}}(\xi) \triangleq \xi \odot \boldsymbol{h}_{\mathbf{X}}(\xi) \in \mathbb{R}^L, \quad (26)$$

under which the symmetric energy decomposes exactly as

$$E_{\mathbf{X}}(\xi) = -\frac{1}{2}\mathbf{1}^{\top}\boldsymbol{\lambda}_{\mathbf{X}}(\xi) = -\frac{1}{2}\sum_{a=1}^{L}[\boldsymbol{\lambda}_{\mathbf{X}}(\xi)]_a. \quad (27)$$

Thus, the scalar value $[\boldsymbol{\lambda}_{\mathbf{X}}(\xi)]_a$ indicates where the local field reinforces the current state ($[\boldsymbol{\lambda}_{\mathbf{X}}(\xi)]_a > 0$) versus where it conflicts with it ($[\boldsymbol{\lambda}_{\mathbf{X}}(\xi)]_a < 0$), providing a direct, spatially resolved view of retrieval stability. We summarize this conflict as the *instability fraction*

$$r_{\mathbf{X}}(\xi) \triangleq \frac{1}{L}\sum_{a=1}^{L}\mathbb{I}([\boldsymbol{\lambda}_{\mathbf{X}}(\xi)]_a < 0). \quad (28)$$

Finally, to quantify the *global* directional agreement between $\xi$ and its induced field, we define the alignment score via cosine similarity, which provides a scale-insensitive summary of whether the retrieved state and its induced field point in a consistent direction:

$$\mathbf{Align_X}(\xi) \triangleq \cos(\xi, \boldsymbol{h}_{\mathbf{X}}(\xi)). \quad (29)$$

Figure 5 provides a schematic summary of these three stability measures.

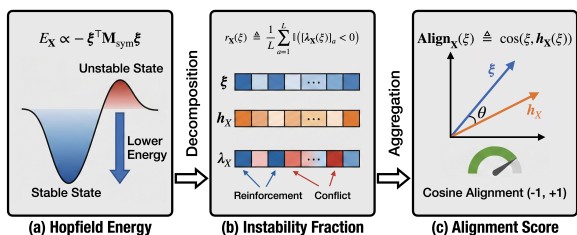

*Figure 5.* **Hopfield-style stability measures.** We characterize the stability of the retrieved state through three complementary lenses: (a) **Hopfield Energy** $E_{\mathbf{X}}$ measuring overall self-consistency; (b) **Instability Fraction** $r_{\mathbf{X}}$ identifying local reinforcement or conflict and (c) **Alignment Score** $\mathbf{Align_X}$ measuring the global directional agreement between the $\xi$ and its induced field.

### 4.2. Retrieval Stability and Perceptual Correlations

Having defined the Hopfield-style stability measures (Equations (21), (28) and (29)), we now examine how these measures relate to externally perceived sample quality and diversity across the generation process.

**Evaluation metrics and protocol.** We compare our Hopfield-style stability measures to three widely used, human-trained metrics that assess distinct dimensions of generation quality: the Aesthetic Score Predictor (Schuhmann et al., 2022) (visual preference), CLIPScore (Hessel et al., 2021) (text–image alignment), and ImageReward (Xu et al., 2023) (preference signals aggregated from curated human feedback). We also report LPIPS (Zhang et al., 2018) diversity as a reference-free proxy for perceptual variation across seeds (Lee et al., 2018). All results use SDXL (Podell et al., 2024) with classifier-free guidance (Ho & Salimans, 2021) $\omega = 5.0$ and 30 sampling steps, generating 1K random-seed samples for each of 10 COCO2014 (Lin et al., 2014) captions (10K total samples).

*Table 2.* **Perceptual quality stratification by Alignment Score.** For each baseline sample, we compute the Alignment Score $\mathbf{Align_X}(\xi)$. We define **Stable/Unstable** regimes as the top/bottom 20% quantiles of $\mathrm{Align_X}(\xi)$ over the full prompt set. For each external metric (ImageReward, AES, CLIPScore), we report the subset mean. The stable subset consistently achieves higher quality scores, whereas the unstable subset shows substantial degradation, suggesting that low stability indicates structural incoherence.

| 1K diverse prompts | $\mathbf{Align_X}$ | ImageReward | Aesthetic | CLIP |
|---|---|---|---|---|
| **1K samples** | 0.669 | 0.546 | 5.643 | 0.263 |
| ▶ **(High-Alignment)** | 0.690 | 0.692 | 5.906 | 0.270 |
| **Top-20% quantile** | (+0.021) | (+0.146) | (+0.263) | (+0.004) |
| ▶ **(Low-Alignment)** | 0.650 | -0.045 | 5.472 | 0.244 |
| **Bottom-20% quantile** | (-0.019) | (-0.591) | (-0.171) | (-0.019) |

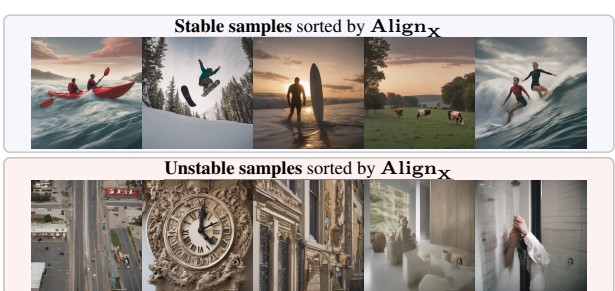

**Stable samples** sorted by $\mathbf{Align_X}$

**Unstable samples** sorted by $\mathbf{Align_X}$

*Figure 6.* **Qualitative visualization of the stability spectrum.** Baseline samples are sorted by their Alignment Score $\mathbf{Align_X}(\xi)$. High-Alignment samples (Stable) exhibit *structural coherence* and consistent object-centric compositions. In contrast, Low-Alignment samples (Unstable) display *fragmented structures* and incompatible texture mixtures, indicating *metastable entrapment*.

**Fidelity–Diversity trade-off via stability measures.** Table 1 shows a consistent association between the proposed stability measures and external evaluation metrics. Stability indicators correlate positively with Aesthetic Score, while showing negative correlations with LPIPS diversity. This suggests that highly stable retrieval is associated with visually coherent generations, whereas lower stability is associated with greater perceptual variation across samples.

The qualitative results in Figure 4 provide a complementary view of this trend. Samples with high $\mathbf{Align_X}(\xi)$ exhibit cleaner structure and fewer hallucinations, but often converge to similar viewpoints or repeated salient features. Conversely, samples with low $\mathbf{Align_X}(\xi)$ show more diverse compositions, while also exhibiting increased structural inconsistencies and artifacts.

**Generalization across diverse prompts.** To validate this relationship under a broader distribution, we extend the analysis to 1,000 COCO2014 captions. Table 2 and Figure 6 confirm that stratifying baseline samples by the Alignment Score $\mathbf{Align_X}(\xi)$ induces consistent shifts in external quality metrics. Specifically, *Stable* (high-alignment) samples exhibit strong structural coherence and object-centricity (often at the expense of diversity), yielding higher perceptual ratings. Conversely, *Unstable* (low-alignment) samples display broader visual variation but suffer from fragmented structures and incoherent feature mixtures, leading to significant quality degradation.

## 5. Methods

Building on the correlation established in Section 4, we propose a training-free mechanism that modulates the attention matrix $\mathbf{QK}^\top$. Our goal is to provide a tunable control over the Hopfield retrieval dynamics, exposing a controllable trade-off between *stability* and *circulation*.

**Modulating circulation via the skew component.** Inspired by classical observations on asymmetric Hopfield networks (Singh et al., 1995; Chengxiang et al., 2000), we utilize the skew-symmetric component of $\mathbf{QK}^\top$ as a lever to control the *circulation dynamics*. This approach is intrinsic to self-attention, since the retrieval operator is constructed from the full attention matrix, which inherently comprises a symmetric part and a skew part:

$$\mathbf{H_X} = \Phi\big(\mathbf{XSX}^\top + \mathbf{XNX}^\top\big). \qquad (30)$$

Since the retrieved features are obtained by applying the retrieval operator to this full matrix and mixing the input features (as in Equation (14)), controlling the skew component provides a direct handle to modulate $\mathbf{H_X}$, thereby influencing the trajectory of the retrieved features $\{\xi^{(i)}\}_{i=1}^{d_{\mathrm{in}}}$ without altering the underlying energy landscape.

### 5.1. Skew Scaling of the Attention Matrix

Recall the classical observation that increasing the asymmetric component leads to an exponential decrease in the total number of stable attractors (Singh et al., 1995). We leverage this property by *scaling* the skew interaction component within the $\mathbf{QK}^\top$. Specifically, we modulate the skew-induced term via a scalar control parameter $\alpha$:

$$\mathbf{XSX}^\top + \mathbf{XNX}^\top \longrightarrow \mathbf{XSX}^\top + \alpha\,\mathbf{XNX}^\top, \qquad (31)$$

which yields the perturbed retrieval operator

$$\begin{aligned} \mathbf{H}_\mathbf{X}^{(\alpha)} &\triangleq \Phi\big(\mathbf{XSX}^\top + \alpha \cdot \mathbf{XNX}^\top\big), \\ \text{yielding} \quad \Xi_\alpha &\triangleq \mathbf{H}_\mathbf{X}^{(\alpha)}\mathbf{X} \in \mathbb{R}^{L \times d_{\mathrm{in}}}, \end{aligned} \qquad (32)$$

where $\mathbf{H}_\mathbf{X}^{(\alpha)}$ denotes the retrieval operator in which the circulation is scaled by $\alpha$.

### 5.2. Blending of Retrieved Features

The circulation-scaled operator $\mathbf{H}_\mathbf{X}^{(\alpha)}$ induces an alternative retrieval state $\Xi_\alpha$ that facilitates perturbation of *metastable mixtures* (Singh et al., 1995; Chengxiang et al., 2000), yet may introduce excessive *state wandering* if the circulation is too strong. To balance these dynamics, we compute the difference vector induced by the perturbation:

$$\Delta \triangleq \Xi_\alpha - \Xi, \qquad (33)$$

and leverage it to form the blended retrieval:

$$\Xi_{\mathrm{blended}} \triangleq \Xi + \beta\,\Delta, \qquad (34)$$

followed by a *normalization step* that matches the baseline feature scale, ensuring that improvements reflect the blending dynamics rather than changes in feature magnitude.

*Table 3.* **Skew-Symmetric Attention Perturbation exhibits an operating-curve on average while selectively repairing failure subsets.** **(a)** MSCOCO–1K reports absolute mean scores over the full prompt set (1K), together with internal Hopfield-style stability measures $(-E_{\mathbf{X}}, r_{\mathbf{X}}, \mathbf{Align}_{\mathbf{X}})$, as we sweep control strengths. **(b)** Low-quality subset blocks report *paired* mean changes $\Delta$ relative to the baseline for the worst 20% baseline samples by each target verifier; gray entries denote side effects on non-target verifiers.

(a) All prompts (avg.).

| Metrics | Baseline | Proposed Methods on SDXL 1K samples | | | | | | | |
|---|---|---|---|---|---|---|---|---|---|
| | | $\alpha = 1.05$ | | | $\alpha = 1.10$ | | | $\alpha = 1.15$ | |
| | | $\beta = 5$ | $\beta = 6$ | $\beta = 7.5$ | $\beta = 4$ | $\beta = 5$ | $\beta = 6$ | $\beta = 3$ | $\beta = 4$ |
| **MSCOCO–1K** | | | | | | | | | |
| Aesthetic Score (↑) | *5.6436* | 5.6657 | 5.6750 | 5.6834 | 5.6971 | 5.7172 | **5.7345** | 5.7042 | 5.7335 |
| ImageReward (↑) | *0.5460* | **0.5756** | 0.5573 | 0.5172 | 0.4992 | 0.4417 | 0.3533 | 0.4449 | 0.3383 |
| CLIPScore (↑) | ***0.2638*** | 0.2632 | 0.2626 | 0.2612 | 0.2605 | 0.2593 | 0.2573 | 0.2597 | 0.2576 |
| $-E_{\mathbf{X}}$ | 3248.17 | 3174.35 | 3153.39 | 3118.93 | 3105.30 | 3060.02 | 2991.55 | 3086.97 | 2989.29 |
| $r_{\mathbf{X}}$ | *0.2314* | 0.2398 | 0.2416 | 0.2443 | 0.2527 | 0.2416 | 0.2453 | 0.2489 | 0.2526 |
| $\mathbf{Align}_{\mathbf{X}}$ | *0.6693* | 0.6540 | 0.6506 | 0.6456 | 0.6504 | 0.6504 | 0.6435 | 0.6366 | 0.6300 |

(b) Low-quality subset (△ values against baseline).

| Metrics | Baseline | Proposed Methods on Low-quality SDXL subset | | | | | | | |
|---|---|---|---|---|---|---|---|---|---|
| | | $\alpha = 1.05$ | | | $\alpha = 1.10$ | | | $\alpha = 1.15$ | |
| | | $\beta = 5$ | $\beta = 6$ | $\beta = 7.5$ | $\beta = 4$ | $\beta = 5$ | $\beta = 6$ | $\beta = 3$ | $\beta = 4$ |
| ▶ **MSCOCO–Low-quality subset**: worst 20% sorted by Aesthetic | | | | | | | | | |
| △ **Aesthetic** | – | **+0.166** | **+0.183** | **+0.243** | **+0.273** | **+0.327** | **+0.331** | **+0.294** | **+0.353** |
| △ ImageReward | – | +0.043 | +0.030 | -0.038 | -0.015 | -0.111 | -0.112 | -0.075 | -0.123 |
| △ CLIPScore | – | +0.004 | +0.001 | -0.001 | -0.002 | -0.005 | -0.006 | -0.002 | -0.009 |
| ▶ **MSCOCO–Low-quality subset**: worst 20% sorted by ImageReward | | | | | | | | | |
| △ Aesthetic | – | +0.022 | +0.004 | +0.017 | +0.012 | +0.040 | +0.006 | +0.018 | +0.010 |
| △ **ImageReward** | – | **+0.453** | **+0.526** | **+0.483** | **+0.518** | **+0.430** | **+0.454** | **+0.450** | **+0.382** |
| △ CLIPScore | – | +0.004 | +0.004 | +0.002 | +0.001 | +0.001 | +0.000 | +0.002 | +0.000 |
| ▶ **MSCOCO–Low-quality subset**: worst 20% sorted by CLIPScore | | | | | | | | | |
| △ Aesthetic | – | +0.019 | +0.004 | -0.013 | -0.015 | +0.039 | +0.035 | +0.001 | +0.044 |
| △ ImageReward | – | +0.116 | +0.099 | -0.005 | +0.026 | -0.043 | -0.073 | -0.051 | -0.079 |
| △ **CLIPScore** | – | **+0.0065** | **+0.0070** | **+0.0075** | **+0.0070** | **+0.0067** | **+0.0072** | **+0.0078** | **+0.0075** |

*Figure 7.* **Qualitative results of feature blending.** *Perturbation on unstable sample* (left): perturbation breaks spurious mixture configurations and yields a cleaner, *object-centric* reconstruction. *Perturbation on stable sample* (right): perturbation injects variation (texture/background/composition) and may introduce drift, illustrating the operating-point trade-off.

Here, $\alpha$ governs the intensity of the circulation perturbation, while $\beta$ regulates the injection of these dynamics into the baseline retrieval. Together, $\alpha$ and $\beta$ provide a controllable trade-off between *stability* and *diversity*.

---

**Algorithm 1** Skew-symmetric perturbation blending

---

**Require:** Input $\mathbf{X}$, association components $\mathbf{M}_{\mathrm{sym/skew}}(\mathbf{X})$
**Require:** Circulation scale $\alpha$, injection scale $\beta$

1: **Input:** Initial query/state $\mathbf{X}$
2: *#1. Standard Retrieval*
3: $\mathbf{A} \leftarrow \mathbf{M}_{\mathrm{sym}}(\mathbf{X}) + \mathbf{M}_{\mathrm{skew}}(\mathbf{X})$
4: $\Xi \leftarrow \Phi(\mathbf{A})\,\mathbf{X}$
5: *#2. Circulation-Scaled Retrieval*
6: $\mathbf{A}_\alpha \leftarrow \mathbf{M}_{\mathrm{sym}}(\mathbf{X}) + \alpha \cdot \mathbf{M}_{\mathrm{skew}}(\mathbf{X})$
7: $\Xi_\alpha \leftarrow \Phi(\mathbf{A}_\alpha)\,\mathbf{X}$
8: *#3. Perturbation via Blending*
9: $\Delta \leftarrow \Xi_\alpha - \Xi$
10: $\Xi_{\mathrm{blended}} \leftarrow \Xi + \beta \cdot \Delta$
11: **Return:** $\Xi_{\mathrm{blended}}$

---

## 6. Results & Discussion

For Table 3 and Figure 7, we apply the proposed method to self-attention retrieval within the UNet by replacing the baseline retrieval $\Xi$ with the blended feature $\Xi_{\mathrm{blended}}$. We follow the experimental protocol in Section 4.2: SDXL with $\omega{=}5.0$ and 30 steps, using the same evaluation metrics. We evaluate on 1,000 COCO2014 prompts (Lin et al., 2014).

**Regime-dependent impact of circulation injection.** Recall that Table 2 and Figure 6 stratified baseline generations by the Alignment Score $\mathbf{Align}_{\mathbf{X}}(\xi)$ into a *Stable* regime and an *Unstable* regime. This separation implies that the utility of circulation injection depends on baseline stability. Therefore, we evaluate whether our method yields the *state-dependent correction* effect suggested by Figure 1: controlled circulation should resolve metastable states while potentially disrupting coherent configurations if excessive.

Table 3b supports this hypothesis through paired, case-conditional evaluation. On the lowest-performing 20% of

*Table 4.* **Stability disruption cost on High-performance baselines.** To complement the rectification results, we evaluate the impact of *circulation injection* on *already high-performing baseline samples*. For each metric, we define a *High-performance subset* by selecting the *top-20% quantile* of baseline samples and report the *paired* mean change on the same prompts.

| Proposed Methods on High-quality SDXL subset | Baseline | $\alpha = 1.05$ | | |
|---|---|---|---|---|
| | | $\beta = 5$ | $\beta = 6$ | $\beta = 7.5$ |
| **MSCOCO–high-performance subset**: top-20% quantile of Aesthetic | | | | |
| $\Delta$ Aesthetic | – | -0.104 | -0.092 | -0.094 |
| **MSCOCO–high-performance subset**: top-20% quantile of ImageReward | | | | |
| $\Delta$ ImageReward | – | -0.096 | -0.131 | -0.190 |
| **MSCOCO–high-performance subset**: top-20% quantile of CLIPScore | | | | |
| $\Delta$ CLIPScore | – | -0.0073 | -0.0095 | -0.0115 |

baseline samples under each metric, the proposed perturbation yields consistent improvements. Qualitatively, Figure 7 illustrates the same regime dependence: on an *Unstable* baseline, circulation injection suppresses incoherent mixture artifacts and produces a cleaner, object-centric reconstruction. In contrast, on a *Stable* baseline, it tends to inject local variation (texture, background, composition) which may lead to unintended deviation.

**Performance trade-offs and cost on high-quality samples.** Aggregate behavior (Table 3a) reflects a trade-off where increasing the circulation parameters $(\alpha, \beta)$ raises Aesthetic Score but can reduce ImageReward and CLIPScore. This state dependence becomes explicit on *High-Performance* baselines. Table 4 reports paired changes on the *top-20% quantile*, revealing that for samples already scoring highly under each external metric, circulation injection produces *degradation* of the corresponding metric. Combined with the substantial gains on the complementary *bottom-20% quantile* (Table 3b), this result suggests that circulation injection perturbs metastable states when baseline retrieval is trapped in poor configurations, yet can disrupt coherent, high-quality configurations when applied excessively.

### 6.1. Operating Regime of Asymmetric Retrieval Dynamics and Adaptive Control

The subset analyses in Tables 3 and 4 suggest that the same circulation perturbation can have different effects depending on the retrieval state. We further interpret this state dependence through the attractor-regime perspective of asymmetric neural networks. As a representative example, Hwang et al. (2019) study deterministic recurrent neural networks and show that the *degree of symmetry* in the connectivity controls the structure of attractors, including fixed points and limit cycles. They quantify this degree of symmetry as

$$\eta_{\mathrm{H}} = \langle J_{ij} J_{ji}\rangle / \langle J_{ij}^2 \rangle, \qquad (35)$$

where $\eta_{\mathrm{H}} = 1$ corresponds to symmetric connectivity, while smaller values indicate increasing asymmetry. In symmetric or near-symmetric regimes, the dynamics are more closely tied to fixed-point-like retrieval, whereas increasing asymmetry can induce cyclic attractors with longer periods. This perspective motivates treating the sym–skew balance not

*Table 5.* **Functional symmetry regimes on SDXL.** We stratify samples by ImageReward and report the realized symmetry index $\eta_M$. Low-performance samples occupy a slightly lower-symmetry regime than average and high-performance samples. Circulation control moves low-performance samples toward this band, but further perturbing already high-performance samples pushes them away from their favorable operating point and reduces quality.

| Skew perturbation | ImageReward ↑ | $\eta_M$ |
|---|---|---|
| **MSCOCO–low-performance subset**: low-20% quantile of ImageReward: | | |
| ✗ | -1.289 | 0.655 |
| ✓ | $-0.819_{\Delta:\ +0.470}$ | 0.659 |
| **MSCOCO–average-performance subset**: avg-20% quantile of ImageReward: | | |
| ✗ | 0.512 | 0.663 |
| **MSCOCO–high-performance subset**: top-20% quantile of ImageReward: | | |
| ✗ | 1.814 | 0.666 |
| ✓ | $1.716_{\Delta:\ -0.098}$ | 0.669 |

merely as a static property of $\mathbf{QK}^\top$, but as an operating parameter that can shift the retrieval dynamics between stable convergence and circulation-driven exploration.

**Functional symmetry of realized attention.** To quantify this operating regime at the sample level, we measure the symmetry–circulation balance of the realized attention interaction. Given the decomposed attention matrix (Equation (18)), we define the functional symmetry index

$$\eta_{\mathbf{M}}(\mathbf{X}) = \frac{\|\mathbf{M}_{\mathrm{sym}}(\mathbf{X})\|_F^2 - \|\mathbf{M}_{\mathrm{skew}}(\mathbf{X})\|_F^2}{\|\mathbf{M}_{\mathrm{sym}}(\mathbf{X})\|_F^2 + \|\mathbf{M}_{\mathrm{skew}}(\mathbf{X})\|_F^2}. \qquad (36)$$

The index is close to 1 when the realized attention interaction is dominated by the symmetric component and decreases (e.g., $\eta_{\mathbf{M}} \to -1$) as the skew-symmetric component becomes stronger. Thus, $\eta_{\mathbf{M}}(\mathbf{X})$ summarizes the relative dominance of energy-supported retrieval and circulation-driven dynamics for the current retrieval state.

**Functional symmetry band.** We next examine whether $\eta_{\mathbf{M}}(\mathbf{X})$ reflects the state-dependent behavior observed in Tables 3 and 4. As shown in Table 5, low-IR samples occupy a slightly lower-symmetry regime than average and high-performing samples. Applying circulation control to the low-performance subset improves IR and moves $\eta_{\mathbf{M}}(\mathbf{X})$ toward the average/high-quality band. However, applying the same perturbation to the high-performance subset increases $\eta_{\mathbf{M}}(\mathbf{X})$ further while decreasing IR.

Thus, the effect of circulation control can be viewed as an under- or over-shift along this operating coordinate: perturbation is beneficial when it moves low-performance retrievals toward the preferred band, but can degrade samples that are already near a favorable regime.

**Adaptive circulation control.** The operating-band behavior above suggests that a fixed $(\alpha, \beta)$ is inherently state-dependent. We therefore consider a lightweight adaptive variant that uses $\eta_{\mathbf{M}}(\mathbf{X})$ to modulate the additional circulation injected at test time. In practice, $\eta_{\mathbf{M}}(\mathbf{X})$ is computed per sample and attention head, and we use a single shared scalar for the corresponding attention call:

$$\bar{\eta}_{\mathbf{M}} \leftarrow \mathrm{Agg}_{b,h}[\eta_{\mathbf{M}}(\mathbf{X})_{b,h}], \qquad (37)$$

*Table 6.* **Adaptive circulation control.** We evaluate a lightweight adaptive variant on 350 COCO samples. For the *moderate* setting, adaptive control preserves the gains of static perturbation across preference metrics while maintaining CLIP and Pick. For the *excessive* setting, static perturbation substantially degrades all metrics, whereas adaptive control recovers from this collapse and improves over the baseline on IR, HPS, and AES.

| Method | IR ↑ | CLIP ↑ | HPS ↑ | AES ↑ | Pick ↑ |
|---|---|---|---|---|---|
| **COCO baseline SDXL subset** | | | | | |
| | 0.487 | 0.264 | 0.2695 | 5.64 | 0.224 |
| **COCO *moderate* skew-perturbation subset**: $(\alpha, \beta) = (1.05, 3)$ | | | | | |
| Static | 0.546 | 0.262 | 0.2730 | 5.66 | 0.224 |
| Adaptive | 0.522 | 0.264 | 0.2723 | 5.64 | 0.224 |
| **COCO *excessive* skew-perturbation subset**: $(\alpha, \beta) = (1.20, 5)$ | | | | | |
| Static | -1.486 | 0.207 | 0.1570 | 5.23 | 0.191 |
| **Adaptive** | **0.568** | **0.264** | **0.2737** | **5.65** | **0.224** |

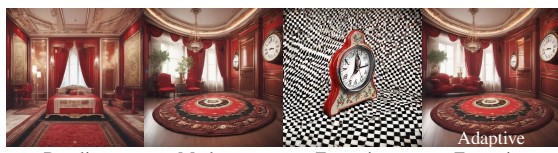

Baseline     Moderate     Excessive     Adaptive Excessive

*Figure 8.* **Effectiveness of adaptive circulation control.** For the prompt "A *fancy clock* ... with red carpet," moderate circulation improves the baseline structure, whereas excessive static circulation introduces visible distortion. Adaptive control reduces this over-perturbation and preserves a more coherent object structure.

where $b$ and $h$ index the sample and attention head, respectively. We then modulate only the deviation from the baseline circulation scale:

$$\alpha_{\text{eff}} \triangleq (\alpha - 1)\bar{\eta}_{\mathbf{M}}. \quad (38)$$

Equivalently, at the logit level, this corresponds to

$$\mathbf{M}_{\text{adap}}(\mathbf{X}) = \mathbf{M}(\mathbf{X}) + \alpha_{\text{eff}} \cdot \mathbf{M}_{\text{skew}}(\mathbf{X}). \quad (39)$$

The adaptive retrieval state is then

$$\boldsymbol{\Xi}_{\text{adap}} = \Phi(\mathbf{M}_{\text{adap}}(\mathbf{X})) \, \mathbf{X}. \quad (40)$$

The blending coefficient $\beta$ controls the step size from the baseline retrieval toward the adaptive retrieval. We therefore use a smaller step when the realized attention is more symmetry-dominated, and a larger step when stronger correction is needed:

$$\beta_{\text{eff}} = \beta(1 - \bar{\eta}_{\mathbf{M}}), \quad (41)$$

and form the final blended retrieval by

$$\boldsymbol{\Xi}_{\text{blend}}^{\text{adap}} = \boldsymbol{\Xi} + \beta_{\text{eff}} \, (\boldsymbol{\Xi}_{\text{adap}} - \boldsymbol{\Xi}). \quad (42)$$

For stability, the implementation additionally matches the feature norm of the blended retrieval to the reference retrieval with a bounded per-token rescaling.

Table 6 and Figure 8 summarize the effect of adaptive circulation control. Table 6 provides quantitative evidence that adaptive control mitigates the degradation caused by excessive static perturbation. Figure 8 gives a qualitative illustration: a moderate static perturbation improves the baseline structure, whereas excessive static perturbation introduces visible distortion. The adaptive variant preserves the intended circulation correction while reducing the over-perturbation caused by the excessive static setting.

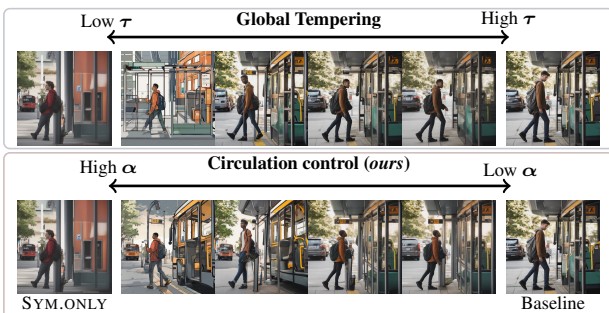

*Figure 9.* **Ablation against attention temperature $\tau$ scaling.** Relative to the SYM.ONLY reference, temperature scaling can introduce unintended structures (e.g., additional leg) due to non-selective strengthening/weakening of interactions across the scene. Instead, our control better preserves strongly supported structure while suppressing weakly supported mixture artifacts.

### 6.2. Circulation Control and Global Tempering

Having characterized the state-dependent operating regime of circulation control, we next validate whether a simpler global attention manipulation can reproduce the same behavior. To this end, we compare our circulation-based perturbation against an attention temperature baseline that linearly rescales the $\mathbf{QK}^\top$:

$$\mathbf{QK}^\top \mapsto \mathbf{QK}^\top / \tau, \quad (43)$$

which globally alters the concentration of the attention distribution. As illustrated in Figure 9, this global modification tends to over-sharpen or under-damp interactions that were already well-structured, producing unintended artifacts (e.g., duplicated limbs). In contrast, our approach acts as a *metastable perturbation*: it tends to preserve the dominant structural support governed by the symmetric component $\mathbf{M}_{\text{sym}}$, while leveraging the skew-symmetric component to suppress weakly supported mixture artifacts, producing more coherent refinements than global temperature scaling at comparable intervention strengths.

## 7. Conclusion, Implications, and Future work

In this work, we propose an associative-memory framework for interpreting self-attention through the structure of $\mathbf{QK}^\top$. By viewing $\mathbf{QK}^\top$ as an association matrix and decomposing it into symmetric and skew components, we derive Hopfield-style stability measures and relate them to the retrieval behavior observed during generation. We further introduce a training-free circulation control mechanism that modulates the skew component using the realized symmetry of attention.

**Implications and Future work.** Our results suggest a complementary way to analyze attention: not only as a token-mixing operator, but also as an interaction matrix with energy-supported and circulation-driven components. This perspective may provide a useful lens for studying attention dynamics beyond diffusion models, including large language models and other transformer architectures.

## Acknowledgments

This work was partly supported by the National Research Foundation of Korea(NRF) grant funded by the Korea government(MSIT) (RS-2024-00335741), Institute of Information & communications Technology Planning & Evaluation (IITP) grant funded by the Korea government(MSIT) (RS-2025-25442405, Development of a Self-Learning World Model-Based AGI System for Hyperspectral Imaging), and Culture, Sports and Tourism R&D Program through the Korea Creative Content Agency grant funded by the Ministry of Culture, Sports and Tourism(RS-2024-00345025, International Collaborative Research and Global Talent Development for the Development of Copyright Management and Protection Technologies for Generative AI).

## Impact Statement

This work offers a principled way to diagnose and mitigate spurious feature mixing in attention-based diffusion models, which may improve the reliability and controllability of generative systems. While the method is broadly applicable to image synthesis, it could also increase the fidelity of generated content in ways that may be misused.

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

# A. Reproducibility and Implementation Details

**Base generator.** All experiments in this paper are conducted using *Stable Diffusion XL (SDXL)* (Podell et al., 2024) with classifier-free guidance (Ho & Salimans, 2021) at a guidance weight of $\omega = 5.0$ and 30 sampling steps.

**Implementation of Skew-symmetric perturbation blending.** During the sampling process, we implement the proposed *circulation-based blending* by intervening on the self-attention retrieval within the UNet layers. Specifically, we replace the baseline retrieval states $\Xi$ with the modulated states $\Xi_{\text{blended}}$ as defined in Equation (34). This intervention is applied globally across the UNet architecture to maintain consistency in the resulting feature trajectories.

**Compute and infrastructure.** Inference is performed on a single NVIDIA GeForce RTX 4090 GPU using 16-bit floating-point (fp16) precision. The experimental framework is implemented using PyTorch (Paszke et al., 2019) and the Hugging Face `diffusers` library (von Platen et al., 2022).

## A.1. Code Implementation

---

**Algorithm 2** Code: Skew-symmetric perturbation blending

---

```python
def get_attn_probs(x_q, W, x_k):
    logits = torch.einsum("bti,hij,bsj->bhts", x_q, W, x_k)
    logits = logits * (1.0 / math.sqrt(d))
    attn_probs = torch.softmax(logits, dim=-1)
    return attn_probs

x_q/k/v = hidden_states
Wq/k/v = attn.to_q/k/v.weight

Cq/k = Wq.shape[1]
H = attn.heads
d = Wq.shape[0] // H

Wq/k_h = Wq.view(H, d, Cq/k)

A_h = torch.einsum("hdi,hdj->hij", Wq_h, Wk_h)
S = 0.5 * (A_h + A_h.transpose(-2, -1))
N = 0.5 * (A_h - A_h.transpose(-2, -1))

attn_probs = get_attn_probs(x_q, (S + self.alpha * N), x_k)
attn_probs_org = get_attn_probs(x_q, (S + N), x_k)

Hx     = torch.einsum("bhts,bsc->bhtc", attn_probs, x_v)
Hx_org = torch.einsum("bhts,bsc->bhtc", attn_probs_org, x_v)

beta, eps, r_min, r_max = self.beta, 1e-6, 0.25, 4.0

Hx_new = Hx_org + beta * (Hx - Hx_org)

ref = torch.linalg.vector_norm(Hx, dim=-1, keepdim=True).clamp_min(eps)
cur = torch.linalg.vector_norm(Hx_new, dim=-1, keepdim=True).clamp_min(eps)
ratio = (ref / cur).clamp(r_min, r_max)

Hx = (Hx_new * ratio).to(dtype=Hx.dtype)

Cv = Wv.shape[1]
Wv_h = Wv.view(H, d, Cv)
hidden_states = torch.einsum("bhtc,hdc->bhtd", Hx, Wv_h)
```

---

# B. Broader Experiments

In this section, we provide additional experiments that examine the broader applicability of the proposed symmetric/skew decomposition and circulation-based control. We first evaluate whether the method transfers to a transformer-based diffusion architecture, and then report a larger-scale COCO evaluation with distribution-level metrics.

## B.1. Generalization to DiT Architectures

We evaluate the proposed circulation control on Stable Diffusion 3, a transformer-based MMDiT architecture (Esser et al., 2024), which follows the broader family of diffusion transformers (Peebles & Xie, 2023). As shown in Table 7, the method transfers beyond the SDXL UNet setting. On the full COCO–1K set, several operating points improve ImageReward (Xu et al., 2023) while keeping Aesthetic Score (Schuhmann et al., 2022) broadly comparable to the baseline. On low-quality subsets, the intervention shows the same regime-dependent pattern observed in the UNet experiments: it improves the target metric, while several settings also yield small or non-negative changes on the other verifiers. These results suggest that the symmetric/skew decomposition and circulation-based control remain meaningful in transformer-based diffusion backbones.

*Table 7.* **Generalization to SD3 MMDiT.** Full COCO–1K reports absolute scores on Stable Diffusion 3. Low-quality subset blocks report paired changes $\Delta$ relative to the baseline.

| Metric | Baseline | $(0.95, 3)$ | $(0.97, 4)$ | $(0.97, 2)$ |
|---|---|---|---|---|
| **(a) Full COCO–1K evaluation on SD3 (Esser et al., 2024)** | | | | |
| IR ↑ | 0.862 | 0.870 | **0.872** | 0.861 |
| AES ↑ | 5.279 | 5.276 | 5.277 | **5.281** |

| Metric | $(0.90, 3)$ | $(0.95, 3)$ | $(0.97, 4)$ | $(0.97, 2)$ |
|---|---|---|---|---|
| ► **Low-quality subset**: bottom-20% sorted by ImageReward | | | | |
| $\Delta$IR ↑ | **+0.505** | **+0.446** | **+0.439** | **+0.229** |
| $\Delta$AES ↑ | -0.049 | +0.018 | +0.011 | +0.006 |
| $\Delta$CLIP ↑ | +0.0043 | +0.0025 | +0.0015 | +0.0025 |
| ► **Low-quality subset**: bottom-20% sorted by Aesthetic | | | | |
| $\Delta$IR ↑ | +0.021 | +0.056 | +0.037 | +0.061 |
| $\Delta$AES ↑ | **+0.224** | **+0.178** | **+0.149** | **+0.146** |
| $\Delta$CLIP ↑ | -0.0049 | +0.0002 | +0.0005 | -0.0000 |

## B.2. Large-Scale Quantitative Evaluation

We also expand the COCO evaluation from the 1K setting to 10K samples and include distribution-level metrics (Heusel et al., 2017; Stein et al., 2023) in addition to the standard perceptual scores. As shown in Table 8, selected operating points improve ImageReward and Aesthetic Score while keeping CLIP close to the baseline. The distribution-level metrics remain broadly comparable to the baseline, indicating that the intervention changes the retrieval behavior without substantially degrading the overall generated distribution. Together with the SD3 results in Table 7, these experiments support the broader applicability of circulation-based attention control across model scale and architecture.

*Table 8.* **COCO–10K quantitative evaluation.** COCO–10K reports standard perceptual scores together with distribution-level metrics. Lower values are better for FID, FD-DINOv2, and KD-DINOv2.

| Method | IR ↑ | AES ↑ | CLIP ↑ | FID ↓ | FD-DINOv2 ↓ | KD-DINOv2 ↓ |
|---|---|---|---|---|---|---|
| **COCO–10K Baseline SDXL (Podell et al., 2024)** | | | | | | |
| | 0.5614 | 5.6321 | **0.2631** | **24.314** | 289.352 | 0.1358 |
| **COCO–10K Ours** | | | | | | |
| $(\alpha, \beta) = (1.03, 3)$ | **0.5753** | 5.6394 | 0.2627 | 24.478 | 289.623 | 0.1358 |
| $(\alpha, \beta) = (1.03, 5)$ | 0.5746 | **5.6432** | 0.2622 | 24.607 | **289.337** | **0.1354** |

## B.3. Qualitative Examples of Success and Failure Cases

To complement the quantitative results, we provide paired qualitative examples in Figure 10. All examples use SDXL (Podell et al., 2024) with the proposed circulation control at $(\alpha, \beta) = (1.05, 3)$.

| **Successful cases** | | **Failure cases** | |
| SDXL | SDXL + Ours | SDXL | SDXL + Ours |

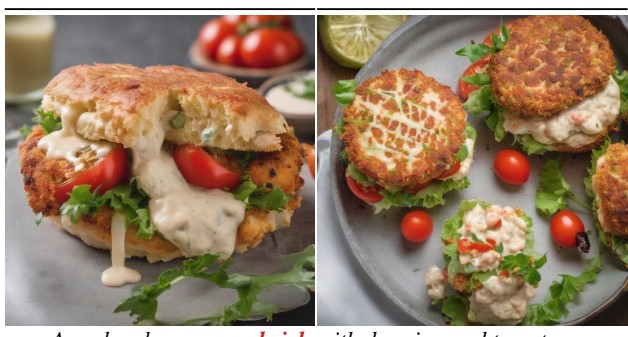

*A laptop with a picture of **the earth** on its screen while sitting on a surfboard.*

*A crab cake on a **sandwich** with dressing and tomatoes.*

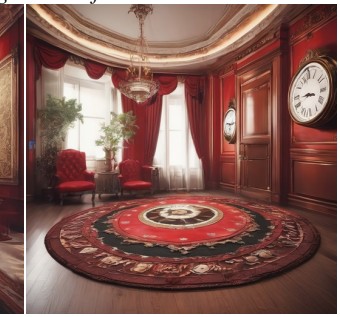
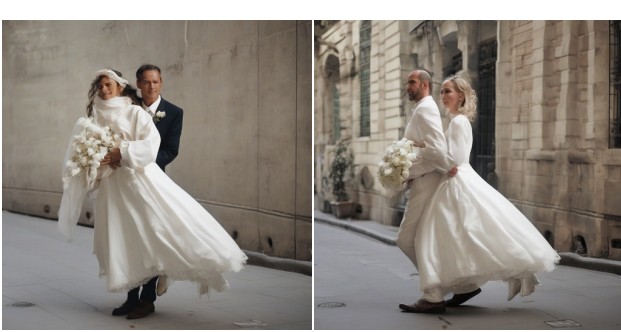

*A **fancy clock** stands in the room with red carpet.*

*A man **carrying** his bride both dressed in white.*

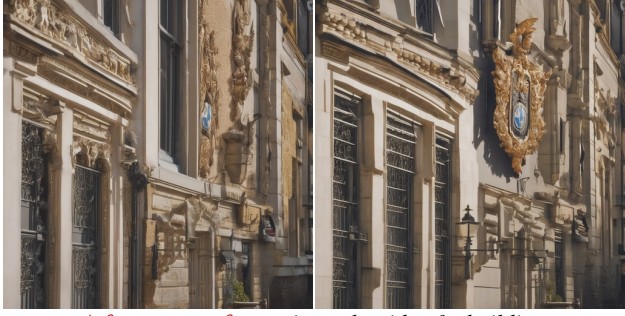
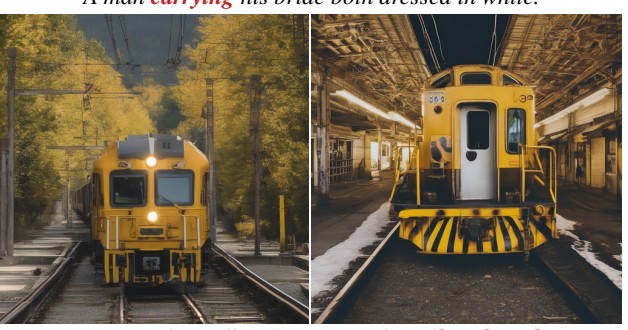

*A **fancy coat of arms** is on the side of a building.*

*A train with a yellow front is on the **railroad tracks**.*

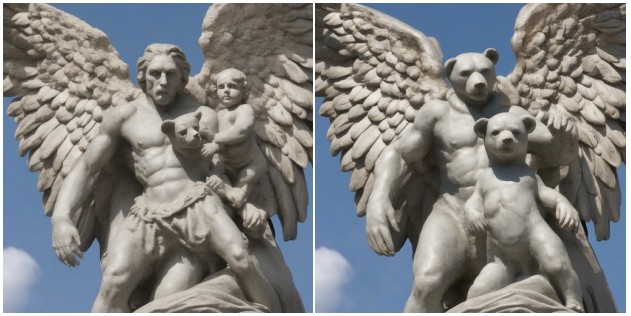
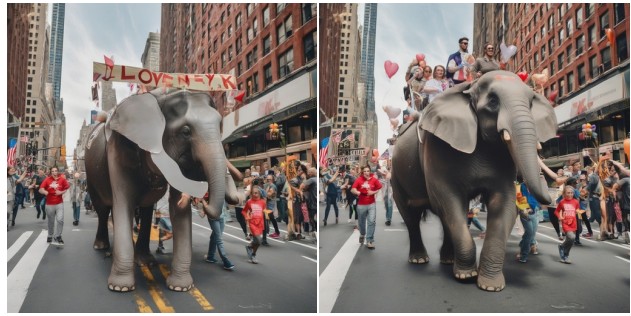

*There are **two stuff bears** on top of an angel statue.*

*A group of people **walking** down the street in a parade with an elephant that says "I love New York".*

*Figure 10.* **Qualitative results of SDXL + Ours**. **Left (successful cases):** the **highlighted** concepts are weakly represented or missing in the baseline, and Ours renders them more faithfully (e.g., adding missing objects or correcting on-screen/scene content). **Right (failure cases):** on prompts the baseline already handles well, Ours can *mildly* degrade the **highlighted** aspect (e.g., the sandwich form, the "carrying" pose, staying on the tracks, or walking vs. riding) while overall image quality remains comparable.

