# OpenReview forum: "Balancing Fidelity and Diversity in Diffusion Models via Symmetric Attention Decomposition: Hopfield Perspective"
_ICML.cc/2026/Conference — ICML 2026 regular_

### Official Review · Reviewer_WV5b · 2026-03-08

**Soundness:** 3
**Presentation:** 2
**Significance:** 3
**Originality:** 3
**Overall Recommendation:** 5
**Confidence:** 4

**Summary:**

The paper “Balancing Fidelity and Diversity in Diffusion Models via Symmetric Attention Decomposition: Hopfield Perspective” offers a novel and robust energy based interpretation of the attention mechanism for image generation. This picture maps the transformation of the embedded input features - which is the core point of transformer models - into a dense version of the Hopfield-like model studied by Singh et al. (1995), i.e. a Hebbian network perturbed by an asymmetric contribution to the coupling matrix. As Singh et al. (1995) relate the degree of asymmetry to a different number of attractors in the attractor landscape, the Authors of the current manuscript exploit this information to: 1) provide insights about the  link between quality metrics and the local shape of the landscape, 2) propose a new guidance method that allows to tune realism and diversity in images.

**Compliance With Llm Reviewing Policy:**

Affirmed.

**Key Questions For Authors:**

I will now report questions and comments that I am asking the Authors to address, in order to improve their manuscript and drive towards my evaluation of the manuscript.

Major Issues:

1. It is not very clear from the main text whether a zero instability ratio is possible or not. One could imagine that, by iterating the hopfield retrieval map we could minimize a dense hopfield-like energy and reach a state (i.e. a final internal representation for the input features) that has no unstable directions, on the line of D’Amico & Negri (2024). Yet, according to my comprehension, for each attention layer parameters are learnt in order to generate the optimized internal representation (i.e. the attention vector in Eq. 17) after one go. As far as I understand, the message of the paper is that the optimized parameters do not drive the model to full stabilization, because it needs the right balance between stability (i.e. symmetry) and instability (i.e. asymmetry) in order to generate realistic but also diverse images. Could the authors confirm if my deduction is correct? If yes, they should make this point clear somewhere in the paper. Otherwise, I ask them to comment on this point and clarify .

2. Moreover, if the Authors could define a symmetry index as done, for instance, in “On the number of limit cycles in asymmetric neural networks” by Hwang et al. (2019), would they be able to characterize, after learning, the optimal degree of (a)symmetry for models trained (without skew scaling) with various data-sets?

3. Why shouldn't weights converge, already, towards an optimal balance between stability and instability, i.e. the correct number and structure of metastable states? Parameters $\alpha$ and $\beta$ are inserted, after learning, to improve generation, but I ask Authors to comment on this mismatch between the learned parameters and the effective ones resulting after adjustment of $\alpha$ and $\beta$. By looking at the test loss, that this depend on the generalization performance of the model ?

4. On the line of the previous question, $\tau$ is inserted - in Section 6.1 - after learning, but it is a multiplicative factor to $QK^{\top}$. As a consequence, shouldn't the optimal $\tau$ be already contained in the values of the weights (a bit like the temperature of a Boltzmann Machine, which, in fact, is set to unity without any change in the performance) ?

Minor Issues:

1. Please, specify that A is a reference matrix in eq. (12).
2. Consistently with the definitions of the other parameters, please define $W_v \in \mathbb{R}^{L \times d_v}$.
3. Figure 3 and Figure 5 are never referenced, also you should swap, for a matter of ordering, figure 4 with figure 5 and cite figure 1 before figure 2.
4. What do the authors mean with “early structural stage” in line 262, second column ?
5. Definitions (30) and (31) are repetitions of definitions (20) and (14), they could be avoided. Otherwise justify them here.

**Limitations:**

yes

**Strengths And Weaknesses:**

*Strengths*:

The paper finally proposes a closed-form energy-based interpretation of the attention mechanism and the dynamics which appears, to me, more robust than previous attempts. Specifically, the analysis performed by the Authors overcomes a series of assumptions from previous works: 1) it does not require symmetry in the dynamic picture, and therefore goes beyond the limited Hopfield interpretation to embody the importance of asymmetry - and so the absence of an actual energy function to be minimized - for generation; 2) it does not require keys (or alternatively queries) to live in the same space as values, an assumption that allowed to see generation in terms of the dynamics of a dense associative memory. I personally consider this analysis more insightful than other attempts, such as the “Energy transformer” proposed by Hoover et al. (2023), which artificially modified the model to impose the symmetric energy analogy.

*Weaknesses*:

The paper could be more clear in the exposition, and more ordered in the way pictures and definitions are presented. For instance, some quantities are defined multiple times. Moreover, the paper lacks an introduction about the role of the attention mechanism in the actual generation of an image, also compared to the application in sequential generation of a text string. Authors should explain, at least briefly, what is the correspondent, in the image case, of a sequence of length L given as an input to the model.

---

> ### Author Rebuttal · Authors · 2026-03-30
>
> We sincerely thank WV5b for the exceptionally thorough and insightful review.
>
> ## C1. Message Clarification
> The reviewer's deduction is correct: our message is not that zero instability is achievable or desirable, but that generation quality requires controlled, non-zero instability. We will add an explicit statement of this interpretation.
>
> ## C2. Characterizing symmetry regimes via a symmetry index
> We adopt a symmetry index analogous to Hwang et al. and examine two complementary quantities:
>
> - **Structural symmetry** $\eta_W$: the degree of intrinsic symmetry in the learned interaction weight matrix $W:=W_QW_K^\top$, measuring how far the parameters themselves have moved from symmetry during training.
> - **Functional symmetry** $\eta_M$: the realized symmetry during sampling, where the learned weights are instantiated as the attention map $M(X)=XWX^\top$ and thus additionally modulated by the input.
>
> **Structural symmetry across architectures.** We first characterize $\eta_W$ across four diffusion model families to ask whether models trained on larger or more diverse data converge to greater or lesser structural symmetry.
>
> |Tab.A|$\eta_W$|
> |---|---|
> |SD1.5|0.291|
> |SD2.1|0.226|
> |SDXL|0.149|
> |SD3|0.151|
>
> Consistent with the reviewer's expectation from Issue 1, no model converges to full symmetry ($\eta_W=1$). Instead, all models settle at a moderate structural asymmetry, with later and larger architectures (SDXL, SD3) retaining less symmetry than earlier ones ($\eta_W\approx 0$). This suggests that, in the case of SD-series, as model capacity and training data scale, the learned interaction weights shift toward greater asymmetry.
>
> **Functional symmetry and the near-optimal working regime.** We further probe whether there exists a preferred $\eta_M$ associated with high-quality generation by stratifying SDXL samples under specific objective(e.g., ImageReward) and computing the corresponding functional symmetry.
>
> |Tab.B|IR|$\eta_M$|
> |---|---|---|
> |Low IR|−1.289|0.655|
> |+ ours|−0.819|0.659|
> |Avg. IR|0.512|0.663|
> |High IR|**1.814**|**0.666**|
> |+ ours|1.716|0.669|
>
> Across baselines, $\eta_M$ increases slightly with IR. In addition, our blended subsets are shifted toward a nearby narrow band around $\eta_M \approx 0.66$. Our blending intervention shifts samples toward this regime, which is consistent with the quality improvements reported in Table 3b of the main paper. While the precise operating regime is likely model- and objective-dependent, the presence of such a narrow band provides a plausible explanation for why post-hoc circulation scaling can be effective.
>
> ## C3. Pre-trained $W$ is not the Optimal Balance
> During training, the diffusion model minimizes a global denoising objective averaged over the full data distribution:
>
> $$
> L:=\mathbb E_{x_0,\epsilon,t}[|\epsilon-\epsilon_\theta(x_t,t)|^2_2]
> $$
>
> This objective forces the parameters to accommodate all modes, noise levels, and compositional variations in the dataset simultaneously. The resulting $\eta_W$ is therefore a **population-averaged compromise**— not a solution tailored to any specific test-time input or evaluation criterion. In particular, there is no mechanism within the training objective that would steer $\eta_W$ toward the narrow near-optimal regime identified in Tab.B for a specific quality metric such as ImageReward.
>
> This argument has an important empirical correlate: as shown in Tab.B, our blending intervention moves low-quality samples from $\eta_M\approx 0.655$ toward $0.659$, approaching the high-quality regime at $0.666$. Post-hoc intervention at test time can target this gap without the distributional averaging that makes training-time correction impractical.
>
> ## C4. Is $\tau$ Already Optimal?
>
> For the same reason as C3, $\tau$ is unlikely to be optimal for a specific test-time objective. Since $W$ is learned under a global denoising loss averaged over all timesteps, noise levels, and data modes — so the attention entropy implicitly encoded in $W$ is again a population-averaged compromise, not a value tuned for any particular quality criterion.
>
> Concretely, because $\tau$ acts multiplicatively on the entire $QK^\top$, it sharpens or flattens all associations simultaneously — beneficial and harmful alike. This non-selectivity is qualitatively illustrated in Fig.8: $\tau$ amplifies already-coherent structure alongside the metastable mixtures it was intended to suppress, introducing unintended artifacts.
>
> > **Role of attention in image domain**. In the image setting, the length-$L$ sequence corresponds to a flattened spatial feature map. Specifically, $F\in\mathbb{R}^{H\times W\times C}$ is reshaped into $X\in\mathbb{R}^{L\times C}$ with $L=HW$, and self-attention models interactions among the resulting spatial tokens.
>
> > **Minor**. We will revise the notation for clarity, fix figure order/citations, clarify that the “early structural stage” refers to the UNet Down blocks, and remove redundant definitions.

---

> > ### Author Rebuttal · Reviewer_WV5b · 2026-04-02
> >
> > I thank the Authors for addressing all my questions in detail.
> >
> > I am happy with the replies and I will keep my current score.

---

> > > ### Author Response · Authors · 2026-04-02
> > >
> > > Dear Reviewer WV5b,
> > >
> > > We sincerely thank you for your thoughtful and constructive feedback throughout the review process. Your comments have been very valuable in improving the clarity and completeness of our paper.
> > > We are glad that our responses were able to address your concerns, and we **appreciate your continued positive evaluation** of our work.
> > >
> > > Thank you again for your time and careful review.
> > >
> > > Sincerely, The Authors

---

### Official Review · Reviewer_ffPi · 2026-03-11

**Soundness:** 3
**Presentation:** 3
**Significance:** 3
**Originality:** 3
**Overall Recommendation:** 4
**Confidence:** 2

**Summary:**

The paper proposes to interpret the pre-softmax attention matrix $QK^T$ in diffusion transformers as an associative matrix, decompose it into symmetric and skew-symmetric parts, use the symmetric part to define Hopfield-style stability measures, and use the skew-symmetric part as a test-time control knob for the fidelity-diversity trade-off.

**Compliance With Llm Reviewing Policy:**

Affirmed.

**Final Justification:**

The authors appropriately addressed my concerns, so I increase the score.

**Key Questions For Authors:**

see limitations below

**Limitations:**

1. The aggregate results in Table 3a show that Aesthetic Score improves but ImageReward and CLIPScore consistently decrease as $\alpha$ and $\beta$ increase. The method thus trades one quality dimension for another rather than achieving a clear Pareto improvement.

2. The controllable knob requires manual tuning.

3. Though the authors provide table to demonstrate the correlation between symmetric/skew decomposition and fidelity/diversity, the method is heuristic and the experiments mostly show that the proposed quantities are associated with better or worse generations, not that they are the main causal drivers of those behaviors.

**Strengths And Weaknesses:**

The symmetric/skew decomposition is conceptually interesting, the method is training-free and simple to implement. The authors also involve empirical results to verify different aspects of the story.

---

> ### Author Rebuttal · Authors · 2026-03-31
>
> Dear reviewer ffPi,
>
> Thank you for your detailed and constructive comments, which greatly helped us improve the paper. We address each point below.
>
> ---
>
> ## C1. Average trade-off in aggregate results
>
> We believe this pattern is consistent with the mechanism of our intervention. Stronger perturbation can help escape metastable, low-quality retrievals, but it can also disturb samples that are already coherent. In this sense, the aggregate in Table 3a averages together two qualitatively different effects: correction in unstable cases and unnecessary deviation in stable ones. This is also why Table 3b and Table 4 should be read together: Table 3b shows selective gains on low-quality subsets, while Table 4 shows the corresponding cost on already high-performing subsets.
>
> The practical implication is that a single static global knob is inherently limited when the effect is regime-dependent.
>
> ## C2. Manual tuning of the control knob
>
> Motivated by the regime dependence discussed in C1, we implemented a preliminary adaptive variant to reduce post-hoc manual tuning. Inspired by the discussion in Reviewer WV5b’s C2 and the related intuition of Hwang et al. (2019), we summarize the realized balance between the symmetric and skew parts of the retrieval interaction by
>
> $$
> s_M(X)=\frac{|M_{\mathrm{sym}}(X)|^2_F-|M_{\mathrm{skew}}(X)|^2_F}{|M_{\mathrm{sym}}(X)|^2_F+|M_{\mathrm{skew}}(X)|^2_F}.
> $$
>
> Intuitively, larger $s_M$ indicates a more symmetry-dominated, energy-forming regime, whereas smaller $s_M$ indicates relatively stronger skew-driven circulation.
>
> Let the additional skew gain induced by $\alpha$ be:
>
> $$
> \delta_sM_{\rm skew}:=(\alpha-1)M_{\rm skew}.
> $$
>
> Then the static intervention is
>
> $$
> M_{\rm static}\leftarrow M+\delta_sM_{\rm skew},
> $$
>
> while the adaptive version modulates only this additional skew gain:
>
> $$
> M_{\rm adap}\leftarrow M+ s_M(X)\delta_s M_{\rm skew},\quad\text{where}\quad\alpha_{eff}:=s_M(X)\delta_s
> $$
>
> In this way, the method shrinks the perturbation back toward the baseline when the current interaction is already relatively asymmetric, instead of injecting additional skew into a state that is already instability-prone.
>
> For $\beta$, we use
>
> $$
> \beta_{eff}:=\beta(1-s_M(X)).
> $$
>
> Here, $\beta$ serves as a step-size modulator for our blending update.
>
> When the realized attention map is highly symmetric $s_M \to 1$, the retrieval is already in a more stable regime and requires little correction, so $\beta_{eff} \to 0$. When $s_M$ is low (high asymmetry), a larger step size permits stronger correction. In this way, $\alpha$ controls the magnitude of the additional skew perturbation, while $\beta$ controls the size of the blending step; the two therefore play complementary roles in the retrieval update.
>
> |(α, β), COCO 350 samples|Method|IR|CLIP|HPS|AES|Pick|
> |-|-|-|-|-|-|-|
> ||Baseline|+0.487|.264|.2695|5.64|.224|
> |(1.05, 3)|$\alpha,\beta$ (static)|+0.546|.262|.2730|5.66|.224|
> ||$\alpha_{eff},\beta$|+0.550|.264|.2723|5.64|.224|
> ||$\alpha_{eff},\beta_{eff}$|+0.522|.264|.2702|5.64|.224|
> |(1.1, 5)|$\alpha,\beta$|+0.421|.259|.2715|5.70|.222|
> ||$\alpha_{eff},\beta$|+0.548|.262|.2747|5.68|.224|
> ||$\alpha_{eff},\beta_{eff}$|**+0.552**|**.263**|**.2721**|**5.65**|**.224**|
> |(1.2, 5)|$\alpha,\beta$|-1.486|.207|.1570|5.23|.191|
> ||$\alpha_{eff}$|+0.391|.259|.2712|5.73|.221|
> ||$\alpha_{eff},\beta_{eff}$|**+0.568**|**.264**|**.2737**|**5.65**|**.224**|
>
> These preliminary results suggest that the adaptive rule has limited effect when the base perturbation is already mild. As the fixed intervention becomes more aggressive, however, the adaptive variants become increasingly beneficial, improving multiple metrics and mitigating the brittleness of static parameter choices; in particular, they help preserve performance when the fixed intervention begins to over-perturb the retrieval dynamics.
>
> |(α, β)|Method|ΔIR|ΔCLIP|ΔHPS|ΔAES|ΔPick|
> |-|-|-|-|-|-|-|
> |(1.1,5)|$\alpha_{eff},\beta$|+.127**|+.002*|+.003**|-.027|+.002**|
> ||$\alpha_{eff},\beta_{eff}$|+.131**|+.004**|+.001|-.055**|+.002**|
> |(1.2,5)|$\alpha_{eff},\beta$|+1.878**|+.052**|+.114**|+.501**|+.030**|
> ||$\alpha_{eff},\beta_{eff}$|+2.054**|+.057**|+.117**|+.418**|+.034**|
>
> ** indicates $p<0.01$ under the paired t-test, and * indicates $p<0.05$. The paired comparisons further support this interpretation. We will extend these experiments and include the finalized results in the appendix.
>
> ## C3. Correlation vs. Causality
>
> We agree that this does not establish formal causality. Our contribution is instead to provide an **interpretable, physics-motivated test-time intervention** grounded in meaningful internal correlations. A tighter causal account of how such internal metrics shape generation dynamics remains important future work, and we view the present study as a step toward that direction.
>
> ---
>
> We appreciate the time and care the reviewer has devoted to reviewing our work, and we hope these responses and revisions adequately address the raised concerns.

---

> > ### Author Rebuttal · Reviewer_ffPi · 2026-04-03
> >
> > My concerns have been adequately addressed.

---

> > > ### Author Response · Authors · 2026-04-03
> > >
> > > Dear Reviewer ffPi,
> > >
> > > We sincerely appreciate your thoughtful comments and positive assessment. We are **grateful that our responses helped address your concerns**, and we **truly appreciate the time and care you devoted to reading our paper and engaging with the discussion**.
> > >
> > > We likewise appreciate your constructive feedback throughout the review process, which helped us improve the paper substantially.
> > >
> > > Sincerely,
> > > The Authors

---

### Official Review · Reviewer_QjEV · 2026-03-11

**Soundness:** 2
**Presentation:** 2
**Significance:** 3
**Originality:** 3
**Overall Recommendation:** 4
**Confidence:** 3

**Summary:**

This paper proposes a novel approach from the perspective of Hopfield associative memory networks , aimed at addressing the fidelity-diversity trade-off and the issue of "spurious mixing" of incompatible features faced by diffusion models in image generation. The authors innovatively decompose the transformer's self-attention matrix into a symmetric component (the energy landscape) that provides structural stability, and a skew-symmetric component (circulation) that drives perturbation. Based on this, they derive quantifiable, Hopfield-style local stability measures. This research introduces a training-free, inference-time intervention mechanism that allows users to inject controlled perturbation by directly modulating the weight of the skew-symmetric component. Experiments demonstrate that this controllable knob can pull the model out of erroneous metastable states without disrupting the underlying structure of high-quality samples , thereby restoring the structural coherence of low-quality generated images.

**Compliance With Llm Reviewing Policy:**

Affirmed.

**Final Justification:**

Thank you for the author's response. I will maintain my score.

**Key Questions For Authors:**

[Generalization to DiT Architectures] Current diffusion models are rapidly evolving towards pure DiT (Diffusion Transformer) architectures based on transformer blocks, which similarly rely heavily on the attention mechanism. Could the authors supplement relevant experiments on such architectures to verify the generalizability of this attention decomposition strategy?
[Quantitative Evaluation on Large-Scale Datasets] Could the authors provide FID scores on a larger-scale dataset (e.g., MS-COCO 30K)? This is crucial to dispel concerns that this mechanism might lead to a degradation of the overall generative distribution.
[Richer Qualitative Demonstrations] It is recommended that the authors supplement the appendix with more comparative visualizations of the generation results across different scenarios and prompts (including successfully repaired cases as well as cases with unintended artifacts or side effects) to more comprehensively demonstrate the actual performance of the proposed method.

**Limitations:**

Yes

**Strengths And Weaknesses:**

Strengths
[Training-Free] This method does not require any expensive model fine-tuning or alignment training. As a plug-and-play inference-time intervention mechanism, it holds high practical engineering value.
[Novel Perspective] The paper connects the attention mechanism with the energy landscape and circulation dynamics in Hopfield associative memory networks , providing a novel physics-based perspective for understanding and controlling feature mixing in diffusion models.
[Ability to Repair Low-Quality Images] Experiments show that this perturbation mechanism can effectively perturb metastable states generated by the model , restoring structural coherence and repairing low-quality generated images that suffer from fragmented structures or incompatible feature mixtures.
Weaknesses
[Lack of Comprehensive Large-Scale Quantitative Evaluation] The paper's current quantitative evaluation is limited to a small-scale subset of 1000 images, which is a relatively small amount of data. For text-to-image diffusion models, there is a lack of widely recognized evaluation metrics such as Zero-shot FID. This makes it difficult for readers to fully gauge the impact of this global intervention mechanism on the model's overall generative distribution.
[Limitations of the Post-hoc Remediation Strategy] This method is essentially a static intervention strategy. As shown in the paper (Table 4) , applying this perturbation to baseline images that already exhibit high generation quality can disrupt their coherent configurations and lead to performance degradation. Because a static parameter selection is suboptimal, the lack of an adaptive trigger mechanism limits the application potential of this method in real-world scenarios.
[Related Work Needs Expansion] This paper focuses on improving the generation performance of diffusion models based on the attention mechanism. Some recent works (e.g., Paper [1]) have optimized the attention mechanism of diffusion models from perspectives such as human intuition. It is recommended to add these to the related works section.
[1] EDT: An efficient diffusion transformer framework inspired by human-like sketching, NeurIPS 24.

---

> ### Author Rebuttal · Authors · 2026-03-31
>
> Dear Reviewer QjEV,
>
> Thank you for your detailed and constructive feedback, which has helped us meaningfully improve both the evaluation and exposition of the paper. We address each point below.
>
> ---
>
> ### **C1. Generalization to DiT Architectures**
>
> We evaluated our method on Stable Diffusion 3, a transformer-based **MMDiT architecture**, using the same evaluation protocol as in Table 3. The results are summarized below.
>
> **Table C1a.** Full COCO-1K evaluation on SD3.
>
> | SD3 | **Full COCO-1K set** | (0.95, 3) | (0.97, 4) | (0.97, 2) |
> |-|-|-|-|-|
> | IR ↑ | 0.862 | 0.870 | **0.872** | 0.861 |
> | AES ↑ | 5.279 | 5.276 | 5.277 | **5.281** |
>
> **Table C1b.** Low-quality subset evaluation on SD3 (∆ relative to baseline).
>
> |Baseline (Low **IR** set)|(0.90, 3)|(0.95, 3)|(0.97, 4)|(0.97, 2)|
> |-|-|-|-|-|
> |ΔIR ↑|**+0.505**|**+0.446**|**+0.439**|**+0.229**|
> |ΔAES ↑|-0.049|+0.018|+0.011|+0.006|
> |ΔClip ↑|+0.0043|+0.0025|+0.0015|+0.0025|
>
> | Baseline (Low **AES** set)|(0.90, 3)|(0.95, 3)|(0.97, 4)|(0.97, 2)|
> |-|-|-|-|-|
> |ΔIR ↑ | +0.021 | +0.056 | +0.037 | +0.061 |
> |ΔAES ↑|**+0.224** | **+0.178** | **+0.149** | **+0.146**|
> |ΔClip ↑|-0.0049|+0.0002|+0.0005|-0.0000|
>
> On the full COCO-1K set, several operating points improve ImageReward while keeping Aesthetic Score broadly comparable to the baseline. On the low-quality subsets, the intervention again shows the same regime-dependent pattern observed in UNet-based model(SDXL): it consistently improves the target metric, while several settings also yield small or non-negative changes on the other verifiers.
>
> Taken together, these results suggest that the proposed symmetric/skew decomposition and circulation-based control remain meaningful beyond the SDXL UNet setting, and can transfer to a transformer-based diffusion architecture as well. We will include the full experimental details and a broader hyperparameter sweep in the revision.
>
> ### **C2. Large-Scale Quantitative Evaluation**
>
> In response to this concern, we expanded the COCO evaluation from 1K to 10K samples and additionally report distribution-level metrics, including FD-Inception (FID), FD-DINOv2, and KD-DINOv2, alongside the standard perceptual scores.
>
> **Table C2.** COCO-10K quantitative results.
>
> ||IR ↑|AES ↑|Clip ↑|FD-Inception (FID) ↓|FD-DINOv2 ↓|KD-DINOv2 ↓|
> |-|-|-|-|-|-|-|
> |Baseline|0.5614|5.6321|**0.2631**|**24.314**|289.352|0.1358|
> |**α,β = 1.03, 3**|**0.5753**|5.6394|0.2627|24.478|289.623|0.1358|
> |**α,β = 1.03, 5**|0.5746|**5.6432**|0.2622|24.607|**289.337**|**0.1354**|
>
> On COCO-10K, the method improves ImageReward and Aesthetic Score at selected operating points, while the distribution-level metrics remain broadly comparable to the baseline and do not indicate substantial degradation of the overall generative distribution. We will further expand this evaluation ($\approx 30K$), including additional sweep configurations, in the revision.
>
> ### **C3. Limitations of the Post-hoc Remediation Strategy**
>
> We kindly ask the reviewer to refer to **our response to Reviewer ffPi (C2)** for the main results. Here, we briefly summarize our results. Here, $X_{eff}$ denotes adaptive variation of $X$.
>
> |(α, β), COCO 350 samples|Method|IR|CLIP|HPS|AES|Pick|
> |-|-|-|-|-|-|-|
> |– | Baseline|+0.487|.264|.2695|5.64|.224|
> |(1.05, 3)|$\alpha,\beta$ (static)|+0.546|.262|.2730|5.66|.224|
> ||$\alpha_{eff},\beta$|+0.550|.264|.2723|5.64|.224|
> ||$\alpha_{eff},\beta_{eff}$|+0.522|.264|.2702|5.64|.224|
> |(1.1, 5)|$\alpha,\beta$|+0.421|.259|.2715|5.70|.222|
> ||$\alpha_{eff},\beta$|+0.548|.262|.2747|5.68|.224|
> ||$\alpha_{eff},\beta_{eff}$|**+0.552**|**.263**|**.2721**|**5.65**|**.224**|
> |(1.2, 5)|$\alpha,\beta$|-1.486|.207|.1570|5.23|.191|
> ||$\alpha_{eff}$|+0.391|.259|.2712|5.73|.221|
> ||$\alpha_{eff},\beta_{eff}$|**+0.568**|**.264**|**.2737**|**5.65**|**.224**|
>
> We will finalize the experiments in the revision.
>
> ### **C4. Related Work Needs Expansion**
>
> We will revise it as follows:
>
> > **Attention mechanisms in diffusion models.**
> >
> > … Oriyad et al. (2025) analyze the entity-missing problem from the perspective of cross-attention maps, and show that excessive overlap between attention maps of different entities is an important contributor to this failure mode.
> >
> > Beyond semantic grounding, recent works have also explored structurally motivated attention priors. **EDT (Chen et al., 2024)** proposes an efficient diffusion transformer framework inspired by human-like sketching, combining a lightweight architecture with a training-free attention modulation matrix and an alternation of local and global attention. …
> >
>
> ### **C5. Richer Qual.**
>
> We will expand the appendix with additional qualitative examples, including successful repairs, failure cases with unintended artifacts at high $\alpha,\beta$, and side-by-side comparisons illustrating the operating-point trade-off.
>
> ---
>
> We appreciate the time and care the reviewer has devoted to reviewing our work, and we hope these responses and revisions adequately address the raised concerns.

---

> > ### Author Rebuttal · Reviewer_QjEV · 2026-04-02
> >
> > Thank you for the author's response. I will maintain my score.

---

> > > ### Author Response · Authors · 2026-04-02
> > >
> > > Dear Reviewer QjEV,
> > >
> > > We sincerely appreciate your thoughtful review and are glad to hear that **all of your concerns have been fully resolved** through our discussion. We are also grateful that you have **maintained your positive perspective** on our work. Your constructive feedback has been invaluable in strengthening our paper.
> > >
> > > Thank you for your time and careful evaluation throughout the review process.
> > >
> > > Sincerely, The Authors

---

### Decision · Program_Chairs · 2026-04-30

**Decision:**

Accept (regular)

**Comment:**

The manuscript contributes to the mechanistic understanding of diffusion transformer models by establishing a link to Hopfield networks. The pre-softmax attention matrix is decomposed into components with play complementary roles in capturing the dynamics of the generation process. Lots of nice visuals empiricall illustrate the results and claims of the paper.

The paper lacks in theoretical grounding. A possible route would be to provide a full theoretical treatement in the case of an analytically tractable setting like mixture of Gaussians. Regardless, I still feel the paper is a good fit for ICLR in its current state.